# Interacting topological quantum chemistry in 2D with many-body real space invariants

Jonah Herzog-Arbeitman [1] ✉, B. Andrei Bernevig [1,2,3] & Zhi-Da Song [1,4,5,6]

The topological phases of non-interacting fermions have been classified by their symmetries, culminating in a modern electronic band theory where wavefunction topology can be obtained from momentum space. Recently, Real Space Invariants (RSIs) have provided a spatially local description of the global momentum space indices. The present work generalizes this real space classification to interacting 2D states. We construct many-body local RSIs as the quantum numbers of a set of symmetry operators on open boundaries, but which are independent of the choice of boundary. Using the $U(1)$ particle number, they yield many-body fragile topological indices, which we use to identify which single-particle fragile states are many-body topological or trivial at weak coupling. To this end, we construct an exactly solvable Hamiltonian with single-particle fragile topology that is adiabatically connected to a trivial state through strong coupling. We then define global many-body RSIs on periodic boundary conditions. They reduce to Chern numbers in the band theory limit, but also identify strongly correlated stable topological phases with no single-particle counterpart. Finally, we show that the many-body local RSIs appear as quantized coefficients of Wen-Zee terms in the topological quantum field theory describing the phase.

The symmetries of a Hamiltonian are essential to the classification of topological phases in crystals. For instance, the Ten-Fold Way[1,2], Topological Quantum Chemistry (TQC)[3,4], and symmetry indicators[5–8] have redefined our understanding of non-interacting electronic states of matter with the symmetry group of the Hamiltonian taking center stage. The success of this program motivates us to extend its reach to interacting Hamiltonians where many-body effects accompany band topology in the groundstate[9–38]. This work focuses on 2D systems with space group $G$ and $U(1)$ charge conservation at an integer filling per unit cell.

The classifications of single-particle topology originally relied on momentum space calculations such as the Wilson loop[39–46] and band structure irreps[7,8,47–50]. Physically, however, nontrivial topology is completely diagnosed in real space where a topological index serves as an obstruction to the atomic limit, defined by a representation of the groundstate with localized, symmetric Wannier states[51–53]. Recently,

ref. 54 extended this idea to symmetry-protected phases by developing Real Space Invariants (RSIs) − local indices which can be considered as Noether charges associated to discrete symmetries − which may be calculated from the irreps formed by the Wannier states in the unit cell, even in fragile phases[55,56]. These RSIs are gauge-invariant and classify all 2D and 3D symmetry eigenvalue-indicated single-particle topology[5,6,57–59], but may not detect some topological states which are classified by cohomology or not protected by symmetry[60–64]. Our paper extends this technique by defining many-body RSIs (henceforth referred to as RSIs for brevity) of two types.

First, we construct local RSIs on open boundary conditions (OBCs) in all 2D point groups. These RSIs classify adiabatically distinct many-body atomic states[9,45,65–68]. We then define many-body fragile topological states[6,9,41,69–72] by an obstruction to all many-body atomic limit states and derive their topological invariants in terms of

[1]Department of Physics, Princeton University, Princeton, NJ 08544, USA. [2]Donostia International Physics Center, P. Manuel de Lardizabal 4, 20018 Donostia-San Sebastian, Spain. [3]IKERBASQUE, Basque Foundation for Science, Bilbao, Spain. [4]Present address: International Center for Quantum Materials, School of Physics, Peking University, Beijing 100871, China. [5]Present address: Hefei National Laboratory, Hefei 230088, China. [6]Present address: Collaborative Innovation Center of Quantum Matter, Beijing 100871, China. ✉e-mail: jonahh@princeton.edu

inequalities between the RSIs and the $U(1)$ particle number. We also present an exactly solvable model verifying that interactions trivialize certain fragile non-interacting states identified by our classification. To study many-body stable topological states[3,73–75], we introduce global RSIs defined on periodic boundary conditions and show that they are many-body stable topological invariants. Finally, we show that the local RSIs appear as quantized coefficients in the topological response theory generalizing the Chern-Simons action of a Chern insulator.

Our theory provides an elementary classification of symmetry-protected Chern and fragile topological phases, provides an explicit connection between many-body and single-particle topological indices substantiated by exactly solvable Hamiltonians, and proposes a fundamental relation between these topological indices defined on the lattice and the topological quantum field theory that describes their universal behavior.

## Results

### Many-body local RSIs

Axiomatically, we define a many-body topological state by an obstruction to adiabatically deforming it into a many-body atomic (trivial) state while respecting the symmetries of the space group $G$. The results of this paper rely on the following definition. A many-body atomic state is any state which is adiabatically connected to a trivial many-body atomic limit which is **(1)** non-degenerate, **(2)** spatially decoupled, and **(3)** endowed with a many-body gap. This limit allows for arbitrarily strong interactions in the Hamiltonian as long as they are strictly local. A many-body atomic limit is the groundstate of the tight-binding Hamiltonian

$$H_{AL} = \sum_{\mathbf{R},\mathbf{r}} H_{\mathbf{R},\mathbf{r}}, \qquad H_{\mathbf{R},\mathbf{r}} = T_{\mathbf{R}} H_{\mathbf{r}} T_{\mathbf{R}}^{\dagger} \tag{1}$$

where $\mathbf{R}$ are the lattice vectors, $T_{\mathbf{R}} \in G$ are the translation operators, $\mathbf{r}$ are the locations of orbitals in the unit cell, and $H_{\mathbf{R},\mathbf{r}}$ is supported only on the orbitals at $\mathbf{R}+\mathbf{r}$ (there are no hoppings). This ensures $[H_{\mathbf{R},\mathbf{r}}, H_{\mathbf{R}',\mathbf{r}'}] = 0$ and thus the groundstate of $H$ can be written as $|GS\rangle = \prod_{\mathbf{R},\mathbf{r}} \mathcal{O}_{\mathbf{R},\mathbf{r}}|0\rangle$ and $\mathcal{O}_{\mathbf{R},\mathbf{r}}$ creates the (possibly correlated) groundstate of $H_{\mathbf{R},\mathbf{r}}$. In the thermodynamic limit, the filling is $\nu = N_{occ}/N_{orb}$ where $N_{orb}$ is the number of orbitals per unit cell and $N_{occ}$ is the filling, e.g. $\prod_{\mathbf{r}} \mathcal{O}_{\mathbf{R},\mathbf{r}}$ creates $N_{occ}$ electrons. Eq. (1) ensures (2) holds, and we require that **(1)** and **(3)** are satisfied, as is natural in an insulator (see Supplementary Note 2).

We now define local RSIs in many-body atomic limits at a Wyckoff position $\mathbf{x}$ protected only by the symmetries of the point group $G_{\mathbf{x}} \in G$. To do so, we identify a set of discrete symmetry operators whose eigenvalues are the local RSIs. This ensures the local RSIs are adiabatic

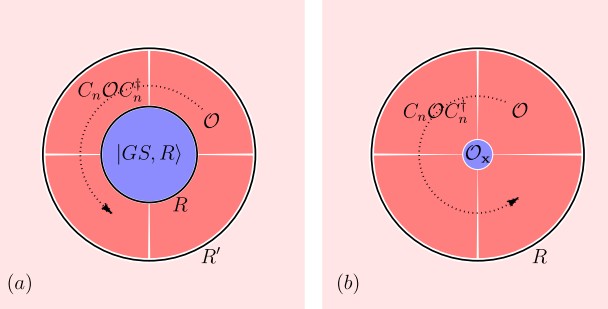

**Fig. 1 | Local quantum numbers. a** We depict the groundstate $|GS,R\rangle$ on OBCs and the additional symmetry-related operators which are included upon expanding the cutoff to $R'$. The many-body local RSIs are invariant under the expansion. **b** Given a fixed cutoff $R$, all operators inside of $R$ but not at the $C_n$-invariant point $\mathbf{x}$ are symmetry-related and so do not contribute to the many-body local RSI. Only the operator $O_{\mathbf{x}}$ at $\mathbf{x}$ transforms locally under $\mathbf{x}$. Its quantum numbers determine the many-body local RSI.

invariants since they are discrete quantum numbers. To ensure locality, we define the local RSI on OBCs by imposing a spatial cutoff around $\mathbf{x}$ and requiring invariance under the particular choice of cutoff.

Let $|GS,R\rangle$ be the groundstate of $H_{AL}$ but restricted to OBCs by including only sites $|\mathbf{R}+\mathbf{r}-\mathbf{x}| \leq R$ in Eq. (1) (see Fig. 1a). The cutoff breaks translational symmetry but preserves the point group symmetries $g \in G_{\mathbf{x}} \subset G$. The quantum numbers of $|GS,R\rangle$ are

$$\hat{N}|GS,R\rangle = N|GS,R\rangle, g|GS,R\rangle = e^{i\lambda[g]}|GS,R\rangle \tag{2}$$

where $\hat{N}$ is the number operator and $e^{i\lambda[g]}$ is a 1D irrep of $G_{\mathbf{x}}$. Note that $|GS,R\rangle$ is a non-degenerate trivial many-body atomic limit and must transform in a 1D irrep **(1)**. However, the quantum numbers $N$ and $\lambda[g]$ are not independent of the cutoff, as we now show, so they cannot be local RSIs. Consider the spinless rotation groups $G_{\mathbf{x}} = n$ generated by the operator $C_n$. Because all terms in $H_{AL}$ are strictly local **(2)**, we can write the groundstate at a larger cutoff $R' > R$ as

$$|GS,R'\rangle = \prod_{i=0}^{n-1} \mathcal{O}_i |GS,R\rangle, \quad \mathcal{O}_i = \prod_{\mathbf{R}+\mathbf{r}\in\mathcal{D}} C_n^i \mathcal{O}_{\mathbf{R},\mathbf{r}} C_n^{\dagger i} \tag{3}$$

where $\mathcal{D}$ is the annulus sector between $R$ and $R'$ of angle $2\pi/n$ shown in Fig. 1a and $C_n \in G_{\mathbf{x}}, C_n^n = +1$ is an $n$-fold rotation. Define the total charge $N_{\mathcal{O}}$ by $[\hat{N}, \mathcal{O}_i] = N_{\mathcal{O}} \mathcal{O}_i$ so that $\mathcal{O}_i \mathcal{O}_j = (-1)^{N_{\mathcal{O}}} \mathcal{O}_j \mathcal{O}_i$. Since the operators $\mathcal{O}_i$ commute/anti-commute if $N_{\mathcal{O}}$ is even/odd, Eq. (3) gives

$$C_n|GS,R'\rangle = e^{i\lambda[C_n]}(-1)^{N_{\mathcal{O}}}|GS,R'\rangle, \quad (n \text{ even}). \tag{4}$$

Thus the $C_n$ eigenvalue $\lambda[C_n]$ is not invariant (for even $n$) under expanding the cutoff because $N_{\mathcal{O}}$ is arbitrary. Similarly, $\hat{N}|GS,R'\rangle = (N+nN_{\mathcal{O}})|GS,R'\rangle$ so $N$ is clearly not invariant. However, we can easily produce symmetry operators which are invariant under an arbitrary expansion of the cutoff. Indeed, $e^{\frac{i\pi}{n}\hat{N}}C_n$ is invariant because

$$\begin{aligned} \left\langle GS,R'|e^{\frac{i\pi}{n}\hat{N}}C_n|GS,R'\right\rangle &= e^{\frac{i\pi}{n}(N+nN_{\mathcal{O}})}(-1)^{N_{\mathcal{O}}}e^{i\lambda[C_n]} \\ 1em &= e^{\frac{i\pi}{n}N}e^{i\lambda[C_n]} = \left\langle GS,R|e^{\frac{i\pi}{n}\hat{N}}C_n|GS,R\right\rangle, \end{aligned} \tag{5}$$

and from Eq. (4), we see immediately that $C_n^2$ is also invariant (for $n$ even). Hence we have found elementary symmetry operators whose eigenvalues only depend on the local properties of the groundstate near $\mathbf{x}$ but are invariant under the imposed cutoff. Their eigenvalues are the local RSIs defined by

$$\begin{aligned} e^{\frac{i\pi}{n}\hat{N}}C_n|GS\rangle &= e^{\frac{i\pi}{n}\Delta_1}|GS\rangle, \quad \Delta_1 \in \mathbb{Z}_{2n} \\ C_n^2|GS\rangle &= e^{\frac{i2\pi}{n/2}\Delta_2}|GS\rangle, \quad \Delta_2 \in \mathbb{Z}_{n/2} \end{aligned} \tag{6}$$

using $(e^{\frac{i\pi}{n}\hat{N}}C_n)^{2n} = (C_n^2)^{n/2} = +1$ which quantizes $\Delta_1, \Delta_2$. Although we derived Eq. (6) in many-body atomic limits, we now prove the local RSIs remain well-defined in general many-body atomic states. First, observe that the operators in Eq. (6) remain symmetries as hoppings and off-site interactions are added to $H_{AL}$ and their eigenvalues (the local RSIs) remain well-defined. Then because we assume a many-body gap **(3)** and the local RSIs are quantized, they cannot change as $H_{AL}$ is adiabatically deformed out of the strict atomic limit.

We have explicitly constructed local RSIs at a single Wyckoff position $\mathbf{x}$ on OBCs. But on infinite boundary conditions, the unit cell contains multiple Wyckoff positions and it should be possible to define local RSIs $\Delta_{\mathbf{x},1}, \Delta_{\mathbf{x},2}$ at each $\mathbf{x}$. For instance in the wallpaper group $G = p2$ which is generated by $C_2$ and translations, there are four Wyckoff positions 1a = (0, 0), 1b = (1/2, 0), 1c = (0, 1/2), and 1d = (1/2, 1/2) whose point groups are generated by $C_2$, $T_1C_2$, $T_2C_2$ and $T_1T_2C_2$ respectively. Even though imposing OBCs at one Wyckoff position breaks the symmetries of the others, we argue that the local RSIs are still well-

defined on infinite boundary conditions. This is because the local RSIs are independent of the cutoff, so sending the cutoff to infinity recovers the full space group by restoring translations.

In Supplementary Note 2, we extend our results to construct local RSIs in all 2D spinless and spinful point groups (see Supplementary Tables 3 and 4). In the spinless groups, we find that mirrors and time-reversal restrict the $C_n$ eigenvalue on the groundstate to be real, reducing the $\mathbb{Z}_{2n} \times \mathbb{Z}_{n/2}$ classification of Eq. (6) to $\mathbb{Z}_{2n}$ for even $n$. For odd $n$, we find a $\mathbb{Z}_n \times \mathbb{Z}_n$ classification which is reduced to $\mathbb{Z}_n$. For even $n$ in the spinful groups, mirrors and time-reversal also force $C_n = \pm 1$ to be real on any non-degenerate state, which yields a $\mathbb{Z}_2$ factor in the local RSI classification. We check that $C_n = +1$ holds on all product states (Slater determinants). The $-1$ eigenvalue is only possible with strong interactions, and can be obtained in trivial atomic Mott insulators'[38,76]. In all cases, the classifying groups are abelian, so the local RSIs are additive: the local RSIs of a tensor product of states is the sum of their individual local RSIs. This is crucial for defining many-body fragile topology.

## Many-body fragile topology

We now consider a state on infinite boundary conditions with charge density $\nu = N_{occ}/N_{orb}$. Recall that single-particle fragile topology is characterized by an obstruction to adiabatic deformation into an atomic state, but this obstruction is removed if additional (trivial) orbitals in a specific representation are added[6]. The topological indices for the single-particle fragile states are inequalities and mod equations relating $N_{occ}$ and the RSIs in the unit cell[54].

This structure extends to the many-body case. We define a topological state with $N_{occ}$ particles per unit cell as many-body fragile iff it can be adiabatically connected to a many-body trivial atomic state with $N_{occ} + \tilde{N}$ particles per unit cell by the addition of $\tilde{N} > 0$ many-body atomic states[9,77]. Let $\Delta_{N_{occ} + \tilde{N}}$ (resp. $\Delta_{\tilde{N}}$) denote the set of RSIs of the trivial state with $N_{occ} + \tilde{N}$ particles (resp. the trivial state of the additional $\tilde{N}$ orbitals) at all Wyckoff positions in the unit cell. The local RSIs of the $N_{occ}$-particle many-body fragile state are defined by $\Delta_{frag} = \Delta_{N_{occ} + \tilde{N}} - \Delta_{\tilde{N}}$ (see Supplementary Note 3 for details). Crucially, $\Delta_{N_{occ} + \tilde{N}}$ is well-defined because the $N_{occ} + \tilde{N}$-particle state is trivial atomic. This underlies the essentially difference between fragile and stable topological many-body states (to be defined shortly), where the latter cannot be connected to any many-body atomic state via the addition of any many-body atomic states[54].

To assess whether a state is topological given a set of local RSIs, we can enumerate all possible atomic limits in $G$ formed from $N_{occ}$ orbitals – note that only a finite number are possible. If the local RSIs of the groundstate do not appear in this set, then the state must be fragile topological by definition. In practice, we find a simpler method by deriving inequality constraints that relate the local RSIs and the $U(1)$ electron density. To illustrate this, we again consider $G = p2$ with Wyckoff positions $\mathbf{x} = 1a, 1b, 1c, 1d$. There are four RSIs given by the eigenvalues $e^{i\frac{\pi}{2}\Delta_{1,\mathbf{x}}}$ of $e^{i\frac{\pi}{2}\hat{N}}C_{2,\mathbf{x}}$ where $C_{2,\mathbf{x}}$ is a rotation centered at $\mathbf{x}$. It is convenient to define $\Delta_{1,\mathbf{x}} \in \{-1, 0, 1, 2\}$. For many-body atomic states on arbitrary OBCs respecting $G_{\mathbf{x}}$, it is easy to prove (see Supplementary Note 3) that $N_{\mathbf{x}} \geq |\Delta_{1,\mathbf{x}}|$ where $N_{\mathbf{x}}$ is the total number of particles. Note that $\Delta_{1\mathbf{x}}$ is does not depend on the OBC cutoff, whereas $N_{\mathbf{x}}$ obviously does. In a many-body atomic limit where we can take $N_{\mathbf{x}} = N_{\mathcal{O}}$ (see Eq. (3)) by choosing a cutoff surrounding $\mathbf{x}$ only (see Fig. 1b), we can bound the total density by summing over the number of states at the high-symmetry Wyckoff positions in a single unit cell:

$$N_{occ} \geq \sum_{\mathbf{x} = 1a, 1b, 1c, 1d} N_{\mathbf{x}} \geq \sum_{\mathbf{x} = 1a, 1b, 1c, 1d} |\Delta_{1,\mathbf{x}}| \quad (7)$$

which is a lower bound because only the high-symmetry Wyckoff positions are counted. The bound in Eq. (7) is ultimately written in terms of RSIs and the charge density which are well-defined quantum

numbers in any many-body atomic state. Eq. (7) holds in all many-body atomic states. Hence if Eq. (7) is violated, then the RSIs impose an obstruction to deformation into a many-body atomic state, proving many-body fragile topology.

We can now prove that certain single-particle fragile states remain fragile topological as interactions are added. As a first step, we determine a formula for the local RSI when acting on product states (which are groundstates without interactions). With $C_2$, the irreps are $A$ and $B$ which are even and odd under $C_2$ respectively. Noting that $e^{i\frac{\pi}{2}\hat{N}}C_2|GS\rangle = e^{i\frac{\pi}{2}(m(A) + m(B)) + i\pi m(B)}|GS\rangle$, Eq. (6) yields

$$\Delta_1 = m(A) - m(B) \bmod 4 \quad (8)$$

where $m(\rho)$ is the multiplicity of the $\rho$ irrep in the product state. In fact, this expression can be understood perturbatively. The single-particle RSI with $C_2$ is $\delta_1 = m(B) - m(A) \in \mathbb{Z}^{54}$. With interactions, a state with $m(A) = 2$ can be scattered into a state with $m(B) = 2$ since both have even parity. Thus states with $\delta_1 = \pm 2$ are identified with interactions[78], and only $\delta_1 \bmod 4 = \Delta_1$ is invariant. Let us now consider the single-particle $N_{occ} = 2$ fragile state $2\Gamma_1 \oplus 2X_1 \oplus 2Y_1 \oplus 2M_2 = (A_{1a} \oplus A_{1b} \oplus A_{1c} \ominus A_{1d})\uparrow G$ which can be thought as stacking a Chern $+1$ state with a Chern $-1$ state. (Here $\uparrow$ denotes the Frobenius induction[47,54] of the irreps $\rho_{\mathbf{x}} \in G_{\mathbf{x}}$ to the full wallpaper group $G$ containing all Wyckoff positions $\mathbf{x}$ as high symmetry points). Although their total Chern number vanishes, there is still fragile topology protected by $C_2$. We compute the RSIs to be $\Delta_{1a} = \Delta_{1b} = \Delta_{1c} = 1$, $\Delta_{1d} = -1$. Evaluating the topological obstruction $\sum_{\mathbf{x}}|\Delta_{\mathbf{x}}| = 4$ in Eq. (7), we find that this single-particle state violates the inequality since $N_{occ} = 2$ and is many-body fragile. Adiabatically adding interactions cannot trivialize the state.

We generalize the inequality criterion of Eq. (7) to all wallpaper groups in Supplementary Note 3, obtaining topological invariants of many-body fragile phases. In Supplementary Table 5, we give expressions for the local RSIs on product states in terms of irrep multiplicities and also single-particle RSIs which are readily computed from momentum space irreps[54]. Our results determine the stability of any single-particle fragile phase when interactions are added.

## Trivializing Single-Particle Fragile Topology

If the local RSIs are compatible with a many-body atomic state, our method indicates there is no obstruction to trivialization even if the single-particle RSIs are nontrivial. We now present an exactly solvable model where we adiabatically deform a single-particle fragile state into a single-particle trivial state through a strong coupling region[64].

Our strategy is to build a non-interacting Hamiltonian with fragile valence bands and obstructed atomic conduction bands. Importantly, we choose the conduction bands to have Wannier functions which are nonzero on a finite number of orbitals[79]. We choose $G = p3$ which has three Wyckoff positions shown in Fig. 2a. Each has spinless PG 3, whose irreps we denote $A$, $^1E$, $^2E$ carrying $C_3$ eigenvalues 1, $e^{i2\pi/3}$, $e^{-i2\pi/3}$ respectively. We create the atomic orbitals $A_{1b}$, $^2E_{1b}$, $^2E_{1c}$ and form the state

$$w^{\dagger}_{0,A_{1a}}|0\rangle = \frac{1}{3}\sum_{j=0}^{2} C_3^j (c^{\dagger}_{0,A_{1b}} + c^{\dagger}_{0,^2E_{1b}} + c^{\dagger}_{0,^2E_{1c}})|0\rangle \quad (9)$$

in the $\mathbf{R} = 0$ unit cell and $w^{\dagger}_{\mathbf{R},A_{1a}} = T_{\mathbf{R}} w^{\dagger}_{0,A_{1a}} T^{\dagger}_{\mathbf{R}} \cdot w^{\dagger}_{0,A_{1a}}$ creates an $A$ irrep at 1a (see Fig. 2a). The complementary two bands on the $A_{1b}$, $^2E_{1b}$, $^2E_{1c}$ orbitals are fragile and cannot be induced from local orbitals (see Supplementary Note 5). We also add the atomic orbitals $^1E_{1a}$ and $^2E_{1a}$ to the valence band (which do not not trivialize the fragile topology) and a $A_{1a}$ orbital to the conduction band. Note that $A_{1a}$ would trivialize the valence band. To construct the non-interacting Hamiltonian $H_0$, we choose all bands to be perfectly flat so $H_0$ is local in the Wannier basis.

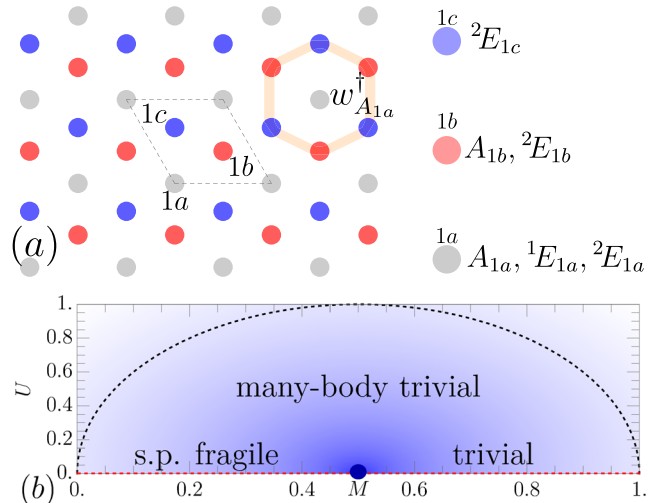

**Fig. 2 | Trivializing fragile topology. a** Orbitals and Wyckoff positions with $w^\dagger_{A_{1a}}$ shown in orange. **b** Phase diagram of $H_0 + H_I$ where shading denotes the many-body gap. The only gapless point (blue) occurs at $M = 1/2$, $U = 0$ separating the single-particle fragile and trivial phases at $U = 0$. Both phases have the same (trivial) many-body RSIs. Along the dashed line, the many-body gap is equal to 1 (see Supplementary Note 5).

As discussed in Supplementary Note 5, we set

$$H_0 = \sum_{\mathbf{R}} (1 - M) w^\dagger_{\mathbf{R}, A_{1a}} w_{\mathbf{R}, A_{1a}}$$
$$1em + M(n_{\mathbf{R}, ^1E_{1a}} + n_{\mathbf{R}, ^2E_{1a}}) + (1 - M) n_{\mathbf{R}, A_{1a}} \tag{10}$$

where $n_{\mathbf{R}, \rho} = c^\dagger_{\mathbf{R}, \rho} c_{\mathbf{R}, \rho}$. All terms in $H_0$ are strictly local because $w_{\mathbf{R}, A_{1a}}$ is finitely supported. At filling $\nu = N_{occ}/N_{orb} = 4/6$, $M$ tunes between a fragile phase:

$$^1E_{1a} \oplus ^2E_{1a} \oplus [A_{1b} \oplus ^2E_{1b} \oplus ^2E_{1c} \ominus A_{1a}] \uparrow G \tag{11}$$

for $M \in (0, 1/2)$ and a trivial phase $A_{1a} \oplus A_{1b} \oplus ^2E_{1b} \oplus ^2E_{1c}$ for $M \in (1/2, 1)$. The two fragile bands in Eq. (11) are in brackets. A gap closing at $M = 1/2$ separates the two phases. However Supplementary Table 5 shows the RSIs are the same in both phases: $\Delta_{1a} = (1, 0)$, $\Delta_{1b} = (2, 1)$, $\Delta_{1c} = (1, 1)$ and indicate a many-body atomic limit (as can be checked using the topological indices in Supplementary Table 8). Accordingly, the single-particle fragile phase can be connected to the trivial phase without a gap closing by adding interactions. We now add the symmetry-preserving term

$$H_I = U \sum_{\mathbf{R}} w^\dagger_{\mathbf{R}, A_{1a}} c^\dagger_{\mathbf{R}, A_{1a}} c_{\mathbf{R}, ^1E_{1a}} c_{\mathbf{R}, ^2E_{1a}} + h.c. \tag{12}$$

which is strictly local. Physically, $H_I$ implements the interaction-allowed conversion[91] $E_{1a} \oplus ^2E_{1a} \to A_{1a} \oplus A_{1a}$ and removes the fragile obstruction symbolized as $\ominus A_{1a}$ in Eq. (11) (but note that $H_I$ annihilates the fragile bands). Because $H_0 + H_I$ acts independently on the Wannier states in each unit cell, the Hamiltonian is entirely decoupled and is trivial to solve[80]. We show the phase diagram in Fig. 2b and see that the gap closing at $U = 0$ separating the single-particle phases can be opened with interactions, adiabatically connecting the phases.

**Many-body stable topology**

Non-interacting stable topological states (such as Chern and quantum spin Hall insulators where many-body invariants are known[81–85]) cannot be trivialized by coupling to any local orbitals – unlike fragile topology. This is reflected in the single-particle RSIs,

which take on fractional values in stable states[54]. For instance with $C_2$, the real-space derivation of the single-particle RSI $\delta_1 = m(A) - m(B) \in \mathbb{Z}$ relies on the existence of Wannier functions such that $m(A), m(B)$ are well-defined integers. It is only by generalizing the definition of $\delta_1$ to momentum space (on periodic boundary conditions) that the possibility of fractional values emerges. In analogy to the non-interacting case, we define a many-body stable topological phase to be robust against coupling to all many-body atomic states. It is impossible to compute local (many-body) RSIs on OBCs in this case because the edge states, a signature of stable topology, prevent our assumption (1) of non-degeneracy and cannot be removed by coupling to any atomic states.

We now propose a definition of global RSIs $\Delta^G_{\mathbf{x}, i}$ in many-body stable topological phases at Wyckoff position $\mathbf{x}$ in the unit cell. Their definition is identical to Eq. (6) but evaluated on a spatial torus, i.e. periodic boundary conditions (PBCs). Explicitly, with $C_n \in G_{\mathbf{x}}$ for $n$ even,

$$e^{\frac{i\pi \hat{N}}{n} C_n} |GS, PBC\rangle = e^{\frac{i\pi}{n} \Delta^G_{\mathbf{x}, 1}} |GS, PBC\rangle,$$
$$C_n^2 |GS, PBC\rangle = e^{\frac{2\pi i}{n/2} \Delta^G_{\mathbf{x}, 2}} |GS, PBC\rangle \tag{13}$$

defines the global RSIs. The PBCs are essential for $|GS, PBC\rangle$ to be non-degenerate so $\Delta^G_{\mathbf{x}, i}$ are quantum numbers. We claim that if $\Delta^G_{\mathbf{x}, i} \neq 0, |GS, PBC\rangle$ is many-body stable topological. In other words, the global RSIs are topological invariants.

To support this claim, we prove two properties of $\Delta^G_{\mathbf{x}, i}$: **(1)** All 2D many-body atomic phases have $\Delta^G_{\mathbf{x}, i} = 0$, from which it follows that all many-body fragile topological phases also have $\Delta^G_{\mathbf{x}, i} = 0$ and **(2)**: $\Delta^G_{\mathbf{x}, i}$ is determined by the Chern number $C$ in non-interacting Slater determinant states. This demonstrates the well-known fact that Chern insulators are robust to weak interactions.

We will first prove **(1)**. Consider a many-body atomic limit $\prod_{\mathbf{R}, \mathbf{r}} \mathcal{O}_{\mathbf{R}, ||0\rangle}$ with $G = p2$ generated by $C_2, T_{\mathbf{R}}$ where $C_2 \in G_{\mathbf{x}}$ is a rotation around the point $\mathbf{x} = (0, 0)$. Now consider placing the state on $L_1 \times L_2$ PBCs with $L_1, L_2$ even. Observe that there are four points invariant under $C_2$ denoted $\mathbf{x}^G = \{(0, 0), (L_1/2, 0), (0, L_2/2), (L_1/2, L_2/2)\}$. Since the $C_2$ operator is a symmetry of each point, it protects a local RSI $\Delta_{\mathbf{x}, 1}$ given by $e^{\frac{i\pi}{2} \Delta_{1, \mathbf{x}}} \mathcal{O}_{\mathbf{x}} = (e^{\frac{i\pi \hat{N}}{2}} C_2) \mathcal{O}_{\mathbf{x}} (e^{\frac{i\pi \hat{N}}{2}} C_2)^\dagger$ for each $\mathbf{x} \in \mathbf{x}^G$. In fact, the $\Delta_{1, \mathbf{x}}$ is the same at each $\mathbf{x} \in \mathbf{x}^G$ because of translations: $\mathcal{O}_{\mathbf{x}} = T_{\mathbf{x}} \mathcal{O}_{(0,0)} T^\dagger_{\mathbf{x}}$ and $C_2 T_{\mathbf{x}} C_2^\dagger = T_{\mathbf{x}}$ since $\mathbf{x} = -\mathbf{x}$ for $\mathbf{x} \in \mathbf{x}^G$ on PBCs.

Now we compute the global RSI of the many-body atomic limit. Using the $C_n$ symmetry, the atomic limit groundstate can generically be written as (see Fig. 3)

$$|GS, PBC\rangle = \prod_{\mathbf{x} \in \mathbf{x}^G} \mathcal{O}_{\mathbf{x}} \prod_{i=1}^2 C_2^i \mathcal{O} C_2^{i\dagger} |0\rangle \tag{14}$$

for some $\mathcal{O}$ which creates the correlated but strictly local groundstates in one half of the spatial torus shown in Fig. 3. The operators $\mathcal{O}_{\mathbf{x}}$ create the states at the corners of the torus which are locally $C_2$ symmetric. Following Eq. (4), we compute

$$e^{\frac{i\pi \hat{N}}{2} C_2} |GS, PBC\rangle = \prod_{\mathbf{x} \in \mathbf{x}^G} e^{\frac{i\pi}{2} \Delta_{\mathbf{x}, 1}} |GS, PBC\rangle \tag{15}$$

which is intuitive because operators $\mathcal{O}$ off the $C_2$ centers contribute trivially. Using Eq. (13), the global RSI of the many-body atomic state is

$$\Delta^G_{\mathbf{x}, 1} = \sum_{\mathbf{x} \in \mathbf{x}^G} \Delta_{\mathbf{x}, 1} = 4\Delta_{\mathbf{x}, 1} = 0 \bmod 4 \tag{16}$$

since we proved that $\Delta_{\mathbf{x}, 1}$ are all equal. The cancelation shown here is due to unexpected coincidence of the global RSI being defined mod 4, and the 4 $C_2$ symmetric points on PBCs each contributing an equal

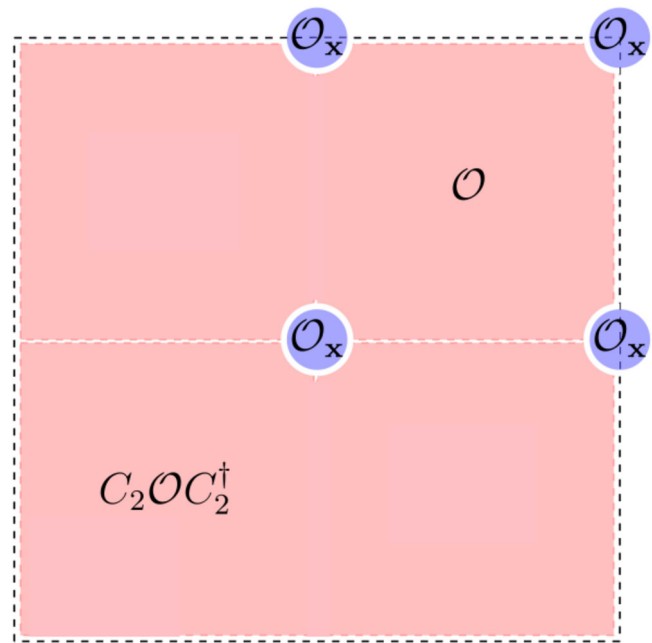

**Fig. 3 | Local Operators on Periodic Boundary Conditions.** We depict the partitioning of a many-body atomic limit state on a spatial torus (PBCs) into rotation-related operators $\mathcal{O}, C_2 \mathcal{O} C_2^\dagger$ and the locally-transforming operators $\mathcal{O}_\mathbf{x}$ operators at fixed points of the rotation $\mathbf{x} \in \mathbf{x}^G$.

(integer) local RSI to the global RSIs. We extend this proof to all point groups in Supplementary Note 4.

Next to prove (2), we relate $\Delta_{\mathbf{x},1}^G$ to the Chern number $C$ in product states. We need two existing results. With $C_2$, $(-1)^C$ is equal to the product of inversion eigenvalues at the high-symmetry points in the Brillouin zone (BZ)[7,28,47,57,86], and secondly $C_2|GS, PBC\rangle = (-1)^C (-1)^{N/2}|GS, PBC\rangle$ where $N$ is the number of states in the BZ which must be even since $L_1, L_2$ are even[87]. Evaluating the global RSI with Eq. (13) yields $\Delta_{1a,1}^G = 2C \bmod 4$. We can also compute $\Delta_{\mathbf{x},1}^G$ at other Wyckoff positions taking e.g. $C_2 \to T_1 C_2$. Then because $|GS, PBC\rangle$ has zero total many-body momentum in 2D, we find

$$\Delta_{1a,1}^G = \Delta_{1b,1}^G = \Delta_{1c,1}^G = \Delta_{1d,1}^G = 2C \bmod 4. \tag{17}$$

This result is reminiscent of the half-integer valued single-particle RSIs in Chern insulators: Eq. (16) gives $\Delta_{\mathbf{x},1}^G = 4\Delta_{\mathbf{x},1}$ when local RSIs are well-defined, so Eq. (17) is suggestive of a half-integer local RSI in odd Chern states.

We compute the global RSIs of all Slater determinants in Supplementary Note 4. Our classification reveals the possibility of many stable topological phases which cannot exist in band theory but are enabled by strong interactions, agreeing with and extending earlier results[78,88,89]. We give a three illustrative examples. In $p2$, a state $|\psi\rangle$ with $\Delta_{1a,1}^G = \Delta_{1d,1}^G = 2, \Delta_{1b,1}^G = \Delta_{1c,1}^G = 0$ obeys $e^{i\frac{\pi}{2}N} C_n|\psi\rangle = -|\psi\rangle$ like in a Chern state, but with nonzero total momentum $T_1|\psi\rangle = T_2|\psi\rangle = -|\psi\rangle$. Such a state carries a many-body Chern number[28], but is adiabatically disconnected from any single-particle Chern state. In $p2mm$, mirrors ensure $C = -C = 0$ without interactions. But the global RSI $\Delta_1^G$ retains a $\mathbb{Z}_4$ classification after adding mirrors, allowing an interaction-enabled state like Eq. (17) following the heuristic $2C = -2C \bmod 4$[90]. Finally, Eq. (17) shows that $\Delta_{\mathbf{x},1}^G$ must be even in gapped Slater determinants. Since an $\Delta_{\mathbf{x},1}^G$ odd requires odd particle number but $L_1, L_2$ are even, this suggests a gapless state. This is evidence that odd $\Delta_{\mathbf{x},1}^G$, defined by Eq. (13), is a many-body semi-metal invariant[91,92].

## Topological response theory

RSIs are quantized, symmetry-protected invariants beyond the Chern number. Then, since a nonzero Chern number is encoded in the continuum topological response theory as a Chern-Simons term[93], it might be expected that RSIs appear as well. In the presence of crystalline symmetries, the response theory includes Wen-Zee-type terms[14,19,31,94–102]:

$$\mathcal{L} = \frac{C}{4\pi} A \, dA + \frac{s}{2\pi} A \, d\omega + \frac{\ell}{4\pi} \omega \, d\omega \tag{18}$$

where $A = A_\mu dx^\mu$ is the $U(1)$ gauge field and $\omega = \omega_\mu dx^\mu$ is the rotational gauge field, or spin connection[10,31,97,103]. Physically, $\int dA$ and $\int d\omega$ are the total flux and total disclination angle. Eq. (18) neglects translational gauge fields[95,104] and hence ignores the unit cell structure, so $\mathcal{L}$ describes an expansion around a fixed Wyckoff position $\mathbf{x}$. We will show that $s$ and $\ell$ are the local RSIs at $\mathbf{x}$.

The coefficients $s$ and $\ell$ can be understood from the equation of motion for the charge density $\rho$ and angular momentum density $L$:

$$\rho = \frac{\delta \mathcal{L}}{\delta A_0} = \frac{C}{2\pi} dA + \frac{s}{2\pi} d\omega, \quad L = \frac{\delta \mathcal{L}}{\delta \omega_0} = \frac{s}{2\pi} dA + \frac{\ell}{2\pi} d\omega. \tag{19}$$

Let us first consider $\rho$. If $d\omega = 0$, Eq. (19) reduces to the Streda formula[105]. If $dA = 0$, $s$ describes the charge bound to a disclination center. A partial disclination with $s \neq 0$ reveals the fractional charge at $\mathbf{x}$[33,98,106,107]. As such, taking $\int d\omega = 2\pi$ to be a complete disclination (in analogy to inserting a full flux), Eq. (19) shows $s = \int \rho$ is the total charge at $\mathbf{x}$. Since the total charge is related to the local RSIs, Eq. (6) gives (for $n$ even)

$$e^{i2\pi s} = e^{i2\pi \hat{N}} = (e^{i\frac{\pi}{n}\hat{N}} C_n)^2 (C_n^2)^\dagger = e^{i\frac{2\pi}{n}(\Delta_{1,\mathbf{x}} - 2\Delta_{2,\mathbf{x}})} \tag{20}$$

acting on $|GS\rangle$ with OBCs. Hence $s$ is the local charge

$$s = \Delta_{1,\mathbf{x}} - 2\Delta_{2,\mathbf{x}} \bmod n \quad (n \text{ even}). \tag{21}$$

We remark that the physical charge bound to the disclination core is a well-defined local observable and can take any (rational) value. However, Eq. (21) shows that its value mod $n$ is determined solely by the many-body RSIs of the defect-less groundstate and is universal. This can be understood from the Lagrangian in Eq. (18): although $\mathcal{L}$ is well-defined for all $s$, adiabatic deformations of the groundstate leave only $s$ mod $n$ constant. In many-body fragile or atomic states where $\Delta_{i,\mathbf{x}}$ are integers, $s \in \mathbb{Z}_n$. Eq. (21) suggests a straightforward generalization to Chern states. Without interactions, the single-particle local RSIs are well-defined but fractional, and a formula for the charge $s$ at $\mathbf{x}$ is known[54,59]. Proving the many-body extension of this result will be the subject of forthcoming work. For now in the $C_2$ case, note that Eq. (16) (proved only at $C = 0$) shows $4\Delta_{\mathbf{x},1} = \Delta_{\mathbf{x},1}^G$, while Eq. (17) shows $\Delta_{\mathbf{x},1}^G = 2C \bmod 4$. Then at least heuristically, the local RSI $\Delta_{\mathbf{x},1} = C/2 \bmod 4$ can be half-integer in agreement with ref. 98.

We now consider $L$ in Eq. (19). Setting $d\omega = 0$ shows that $s$ shifts the angular momentum after inserting a full flux. Indeed, single-particle RSIs can enforce irrep flow due to angular momentum pumping in flux[59,108]. Setting $dA = 0$ shows that $\ell$ describes the angular momentum bound to a disclination center and should be identified with the eigenvalue of the rotation operator $e^{i\frac{\pi}{n}\hat{N}} C_n$. Hence we propose $\ell = \Delta_{1,\mathbf{x}} \in \mathbb{Z}_{2n}$ for $n$ even, matching the classification of ref. 98.

## Discussion

TQC is a unifying theory of non-interacting materials. Given atomic orbitals, their symmetries, and the number of electrons, the topological invariants of TQC classify possible gapped, degenerate, and gapless phases. The present work achieves the first case in interacting

Hamiltonians by defining local and global RSIs, many-body topological indices, and the effective field theory that governs them. In so doing, we revealed which single-particle fragile phases survive interactions and identified undiscovered stable topological states with no single-particle counterpart. We anticipate that many features of single-particle topology may be generalized to the many-body case using this formalism, for instance regarding bounds on quantum geometry[15,109–114]. Another perspective is offered by ref. 28, which shows that a generalization of the single-particle Brillouin zone is the flux torus obtained from twisted boundary conditions (on which the gapped, many-body groundstate can smoothly be defined), since in both cases the Berry curvature and symmetry eigenvalues can be defined. This connection may facilitate the computation of many-body RSIs without open boundary conditions and reveal connections between the momentum space and real space theories. We leave the study of global and magnetic symmetries, spontaneous symmetry breaking, the numerical investigation of interaction-enabled many-body stable topology, and the study of boundary signatures to future work.

## Inclusion and ethics

A portion of this work was completed at Princeton University, which occupies part of the ancient homeland and traditional territory of the Lenape people. We pay respect to Lenape peoples past, present, and future and their continuing presence in the homeland and throughout the Lenape diaspora.

## Data availability

All data the supporting the findings of this study are available form the authors upon reasonable request.

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

## Acknowledgements

We thank Adrian Po for influential comments in the early stages of this work and Abhinav Prem for his insight and careful explanations. Additionally we thank Frank Schindler for useful guidance, Jiabin Yu, and Biao Lian for enlightening consultations, and Titus Neupert, Glenn Wagner, and Martina Soldini for sharing their unpublished paper on many-body atomic limits in the final states of this work. JHA thanks the Donostia International Physics Center for their hospitality during the completion of this manuscript. B.A.B. and Z-D.S. were supported by the European Research Council (ERC) under the European Union's Horizon 2020 research and innovation programme (grant agreement No. 101020833), the ONR Grant No. N00014-20-1-2303, the Schmidt Fund for Innovative Research, Simons Investigator Grant No. 404513, the Packard Foundation, the Gordon and Betty Moore Foundation through the EPiQS Initiative, Grant GBMF11070 and Grant No. GBMF8685 towards the Princeton theory program. Further support was provided by the NSF-MRSEC Grant No. DMR-2011750, BSF Israel US foundation Grant No. 2018226, and the Princeton Global Network Funds. JHA is supported by a Hertz Fellowship and ONR Grant No. N00014-20-1-2303, as well as a Marshall Scholarship during the early stages of this project. Z.-D.S. was supported by the National Key Research and Development Program of China (No. 2021YFA1401903), National Natural Science Foundation of China (General Program No. 12274005), and Innovation Program for Quantum Science and Technology (No. 2021ZD0302403).

## Author contributions

JHA, BAB, and ZDS all contributed to the intellectual content of the work.

## Competing interests

The authors declare no competing interests.
