## [Peer Review File · Nature Communications]

REVIEWER COMMENTS

Reviewer #1 (Remarks to the Author):

In this work authors developed real space invariants for interacting electrons in two dimensions.

They discussed point group symmetries and wallpaper groups in two dimensions with or without spin-orbit coupling (i.e., their theory is applicable to both spinful and spinless electrons). Their invariants detect topological phases such as fragile topological insulators and Chern insulators. Although there is some room for improvement of presentations, I believe the results are useful and worth publishing after revision.

(1) The following two pioneering works on fragile topological phases are missing from the manuscript.

H. C. Po, H. Watanabe, and A. Vishwanath, Phys. Rev. Lett. 121, 126402 (2018).

J. Cano, B. Bradlyn, Z. Wang, L. Elcoro, M. G. Vergniory, C. Felser, M. I. Aroyo, and B. A. Bernevig, Phys. Rev. Lett. 120, 266401 (2018).

(2) I guess the global RSI in Eq 15 is for atomic limit. Although this is stated in the paragraph starting with "Consider a many-body atomic limit" below Eq 12, this is too far and the the current presentation is a bit misleading. Please clarify this point.

(3) Eq 16 states that, for band insulators, the RSI $\Delta_x, 1^G$ is common among the four Wyckoff positions (1a,1b,1c,1d) and the common value is given by two times the Chern number mod 4.

However, the proceeding paragraph contains a more general example with $\Delta_x, 1^G$ is 2 for $x=1a,1d$ while $\Delta_x, 1^G=0$ for $x=1b$ and $1c$. In this case is the Chern number well-defined? If yes, what is how to judge the Chern number based on RSI?

(4) The following work also related the rotation eigenvalues of many-body ground state and the Chern number. I believe a brief comment on the relation is useful to readers.

A. Matsugatani, Y. Ishiguro, K. Shiozaki, and H. Watanabe, Phys. Rev. Lett. 120, 096601 (2018)

(5) Although the analysis of this work is mathematically correct, the main text only contains brief explanations and all important details are hidden in supplementary informations. I believe it is not

easy for readers to understand the logic. Please try to move some of the materials in SI to the main text.

Reviewer #2 (Remarks to the Author):

This work studies interacting topological states in two dimensions from real space representations. It is part of a series of works starting from Ref. 3, and is more directly based on Ref. 53, in which some of the authors of the concerned manuscript developed real space invariants (RSIs) as indicators of noninteracting topological states. In this work, the authors extended the notion by including electron interactions to define many-body RSIs.

While their full results are contained in the appendices, the essential discussion in the main text is mostly understandable without knowing their previous works about topological quantum chemistry and RSIs. The downside, though, is that the description is primarily mathematical, which perhaps hinders readership from those who are not familiar with the classification of topological states of matter. Nevertheless, their analysis seems sound as it is based on their previous publications. Their classification tables in appendices and, particularly, the inequality Eq. (7) would be of potential interest in materials science, so that the manuscript at last may be published in Nature Communications.

Regarding mathematical descriptions, it seems that the work lacks enough explanations. For example, the notation $G_x = n$ in p. 2 is not clear at first sight, though understandable, and particularly an uparrow (\uparrow) requires an explanation. The notation is explained in the appendices while it was used in previous papers such as Refs. 3 and 53, but it requires at least a comment in the main text. In addition, Eq. (7) contains $\Delta_{x$, which I presume $\Delta_{1,x$.

In the section "Topological Response Theory", the authors postulate the values of the coefficients s and l in the Wen-Zee-type terms. They propose the values of s and l from RSIs with modulo n . The postulations seem reasonable, but as s signifies the charge bound to a disclination center, the constraint about modulo n should be explained. It seems that only this section describes the physical consequences of RSIs, so that a careful but definite statement would be useful for readers.

REVIEWER COMMENTS

Reviewer #1

We would like to begin for thanking Reviewer #1 for their careful assessment of our paper. We are pleased to see that they understand the utility and scope of our work, and we are grateful for the chance to improve it with their suggestions. We would also like express gratitude to the reviewer for their expert knowledge of the field, and for pointing out an important work missed in our initial manuscript.

In this work authors developed real space invariants for interacting electrons in two dimensions.

They discussed point group symmetries and wallpaper groups in two dimensions with or without spin-orbit coupling (i.e., their theory is applicable to both spinful and spinless electrons). Their invariants detect topological phases such as fragile topological insulators and Chern insulators. Although there is some room for improvement of presentations, I believe the results are useful and worth publishing after revision.

We have attempted to improve the overall readability of the paper by adding contextualizing remarks and physical interpretations of our mathematical approach. The red/blue in-line edits in the attached manuscript show edits we have made, which we explain in detail now.

(1) The following two pioneering works on fragile topological phases are missing from the manuscript.

H. C. Po, H. Watanabe, and A. Vishwanath, Phys. Rev. Lett. 121, 126402 (2018).

J. Cano, B. Bradlyn, Z. Wang, L. Elcoro, M. G. Vergniory, C. Felser, M. I. Aroyo, and B. A. Bernevig, Phys. Rev. Lett. 120, 266401 (2018).

We thank the reviewer for pointing out these foundational papers, which we have now appropriately included in the introduction.

(2) I guess the global RSI in Eq 15 is for atomic limit. Although this is stated in the paragraph starting with “Consider a many-body atomic limit“ below Eq 12, this is too far and the the current presentation is a bit misleading. Please clarify this point.

Indeed this is correct, the result in Eq. 15 shows the global RSI vanishes in all atomic limit states. We have made the following edits to clarify the logic in this section:

1. After Eq. 12, we indexed our two lemmas (1) and (2) to streamline the argument for the reader.
2. In the proof of lemma (1), we clarify in several instances that we refer to many-body atomic limits.
3. We have added the following explanation “which creates the correlated but strictly local groundstates in one half of the spatial torus (see App. D1). The

operators O_x create the states at the corners of the torus which are locally C_2 symmetric” after Eq 13.”

4. We also summarize proof of (1) with the sentence “The cancelation shown here is due to unexpected coincidence of the global RSI being defined mod 4, and the 4 C_2 symmetric points on PBCs each contributing an equal (integer) local RSI to the global RSIs.”

(3) Eq 16 states that, for band insulators, the RSI $\Delta_{x,1}^G$ is common among the four Wyckoff positions (1a, 1b, 1c, 1d) and the common value is given by two times the Chern number mod 4.

However, the proceeding paragraph contains a more general example with $\Delta_{x,1}^G$ is 2 for $x=1a, 1d$ while $\Delta_{x,1}^G=0$ for $x=1b$ and $1c$. In this case is the Chern number well-defined? If yes, what is how to judge the Chern number based on RSI?

Happily, the central result of *Matsugatani et al.* shows that the state proposed above does carry a Chern number (since the total angular momentum / product of C_2 eigenvalues is nonzero with fermionic minus signs properly accounted for). The generalization of this statement is that the 1a global RSI determines the Chern number mod n for a C_n rotation. We emphasize that this is merely a restating of known results, e.g. *Matsugatani et al.* But it is also generalized by our claim that any nonzero global RSI indicates a strong topological phase: nonzero angular momentum (C_n eigenvalues) or linear momentum (T_R eigenvalues) diagnose symmetry-protected topological phases disconnected from Slater determinants. We have briefly summarized this in the text with the addition “Such a state carries a many-body Chern number ν [Matsugatani et al.] but is adiabatically disconnected from any single-particle Chern state.”

(4) The following work also related the rotation eigenvalues of many-body ground state and the Chern number. I believe a brief comment on the relation is useful to readers. A. Matsugatani, Y. Ishiguro, K. Shiozaki, and H. Watanabe, *Phys. Rev. Lett.* 120, 096601 (2018)

We are grateful to be alerted to this reference, which makes a deep connection between the flux torus used to define the Chern number as a topological index of a many-body states through the Hall conductance and the symmetry eigenvalues of the groundstate at symmetric fluxes. As the authors point out, it is the toroidal flux that most naturally generalizes the single-particle momentum. Although the present work was inspired by a real space perspective on topology, *Matsugatani et al.* provide a natural route for a momentum space generalization: one should compute the “band representation” of the groundstate on the flux torus. This is beyond the scope of the present paper, but it potentially very fruitful and elegant. The insight obtained from this work and independent group cohomology calculations is that such a scheme must recover the reduced classification due to interactions.

In addition to citing the above work in the introduction and the calculation of the many-body symmetry eigenvalues, we have added the following to the conclusion: “Another perspective is offered by \Ref{PhysRevLett.120.096601}, which shows that a generalization of the single-particle Brillouin zone is the flux torus obtained from twisted boundary conditions (on which the gapped, many-body groundstate can smoothly be defined), since in both cases the Berry curvature and symmetry eigenvalues can be defined. This connection may facilitate the computation of many-body RSIs without open boundary conditions and reveal connections between the momentum space and real space theories.”

(5) Although the analysis of this work is mathematically correct, the main text only contains brief explanations and all important details are hidden in supplementary informations. I believe it is not easy for readers to understand the logic. Please try to move some of the materials in SI to the main text.

We appreciate the chance to improve the readability of the manuscript, while acknowledging its integrity and usefulness. In our opinion, the key elements of the argument for the main text are the construction of the local and global RSIs (at least for the C₂ case). To make this argument complete within the space allotted for the main text, we have moved three figures from the appendices to the appropriate sections. We have adapted the text to incorporate these figures into the argument and make more of the visual nature of the proof clear.

Reviewer #2 (Remarks to the Author):

This work studies interacting topological states in two dimensions from real space representations. It is part of a series of works starting from Ref. 3, and is more directly based on Ref. 53, in which some of the authors of the concerned manuscript developed real space invariants (RSIs) as indicators of noninteracting topological states. In this work, the authors extended the notion by including electron interactions to define many-body RSIs.

While their full results are contained in the appendices, the essential discussion in the main text is mostly understandable without knowing their previous works about topological quantum chemistry and RSIs. The downside, though, is that the description is primarily mathematical, which perhaps hinders readership from those who are not familiar with the classification of topological states of matter. Nevertheless, their analysis seems sound as it is based on their previous publications. Their classification tables in appendices and, particularly, the inequality Eq. (7) would be of potential interest in materials science, so that the manuscript at last may be published in Nature Communications.

We thank reviewer #2 for their positive appraisal of the manuscript, and in particular for their appreciation of the fact that the many-body RSIs are suitable for computation in real materials. (This distinguishes our work from more abstract group cohomology

calculations which do not provide practical formulae.) The reviewer has understood the development of this work as a development of topological quantum chemistry to include interaction effects as present in realistic materials. We are pleased that they assess our work can “be published in Nature Communications” once the following edits are made.

Regarding mathematical descriptions, it seems that the work lacks enough explanations. For example, the notation $G_x = n$ in p. 2 is not clear at first sight, though understandable, and particularly an \uparrow requires an explanation. The notation is explained in the appendices while it was used in previous papers such as Refs. 3 and 53, but it requires at least a comment in the main text. In addition, Eq. (7) contains Δ_x , which I presume $\Delta_{1,x}$.

We sympathize with the reviewer, the notation is rather terse. We have addressed each instance raised in the preceding paragraph. We added the sentence “Consider the spinless rotation groups $G_{\mathbf{x}} = n$ generated by the operator C_n .” on page 2 to clarify the meaning of n as the point group, and “For instance in the wallpaper group $G=p2$ which is generated by C_2 and translations” on page 3 to clarify the structure of the group and explain notation. Lastly, we added a definition of the arrow via the sentence “(Here \uparrow denotes the Frobenius induction^{\cite{rsis,2020arXiv200604890C}} of the irreps $\rho_{\mathbf{x}} \in G_{\mathbf{x}}$ to the full wallpaper group G containing all Wyckoff positions \mathbf{x} as high symmetry points.)” with appropriate references for full mathematical details. Luckily, we do not make use of this formalism much in the present work.

Lastly, the reviewer is certainly correct about Eq. 7, and we have corrected it thanks to their keen eyes.

In the section “Topological Response Theory”, the authors postulate the values of the coefficients s and l in the Wen-Zee-type terms. They propose the values of s and l from RSIs with modulo n . The postulations seem reasonable, but as s signifies the charge bound to a disclination center, the constraint about modulo n should be explained. It seems that only this section describes the physical consequences of RSIs, so that a careful but definite statement would be useful for readers.

The reviewer raises a good point about the effective field theory that can be answered concretely using our lattice-oriented approach. The local charge is a bona fide observable and is defined without any modulus. However, symmetry can only provide its value mod n , which is the universal part since adiabatic changes of the groundstate may alter the charge density. However, its mod n remainder must be invariant. We have added the following remarks to indicate this: “We remark that the physical charge bound to the disclination core is a well-defined local observable and can take any (rational) value. However, $\text{Eq}\{eq:sRSIs\}$ shows that its value mod n is determined solely by the many-body RSIs of the defect-less groundstate and is universal. This can be understood from the Lagrangian in $\text{Eq}\{eq:lag\}$: although lag is well-defined for all s , adiabatic deformations of the groundstate leave only $s \pmod n$ constant.”

REVIEWERS' COMMENTS

Reviewer #1 (Remarks to the Author):

In the revised manuscript, the authors addressed my concerns one by one.

Although I still believe that more materials in SI should be moved to the main text for its readability without SI, the authors do not seem to like that idea. I think the revised manuscript may be published from Nature Communications.

Reviewer #2 (Remarks to the Author):

After the revision, the authors improved the manuscript, reflecting the comments from the two referees. I now think that the manuscript is suitable for publication in Nature Communications.